# Rpd3L and Hda1 histone deacetylases facilitate repair of broken forks by promoting sister chromatid cohesion

Pedro Ortega[1], Belén Gómez-González [1]* & Andrés Aguilera [1]*

Genome stability involves accurate replication and DNA repair. Broken replication forks, such as those encountering a nick, lead to double strand breaks (DSBs), which are preferentially repaired by sister-chromatid recombination (SCR). To decipher the role of chromatin in eukaryotic DSB repair, here we analyze a collection of yeast chromatin-modifying mutants using a previously developed system for the molecular analysis of repair of replication-born DSBs by SCR based on a mini-HO site. We confirm the candidates through FLP-based systems based on a mutated version of the FLP flipase that causes nicks on either the leading or lagging DNA strands. We demonstrate that Rpd3L and Hda1 histone deacetylase (HDAC) complexes contribute to the repair of replication-born DSBs by facilitating cohesin loading, with no effect on other types of homology-dependent repair, thus preventing genome instability. We conclude that histone deacetylation favors general sister chromatid cohesion as a necessary step in SCR.

[1] Centro Andaluz de Biología Molecular y Medicina Regenerativa (CABIMER), Universidad de Sevilla-CSIC-Universidad Pablo de Olavide, Seville, Spain. *email: gomezb@us.es; aguilo@us.es

The maintainance of genetic information through cell divisions requires accurate duplication of the entire genome, despite the presence of DNA lesions[1]. Failures during DNA replication can lead to genome instability, most cancer mutations being attributed to replication errors[2]. The most common DNA lesions are single-strand DNA (ssDNA) breaks or nicks[3,4], which when encountered by a replication fork are converted into double-strand breaks (DSBs) that are among the most harmful lesions. Besides nicks, fork stalling at other DNA lesions can also ultimately lead to fork collapse or even DSBs, particularly in the absence of a proper checkpoint-mediated fork stabilization[5].

DSB repair can be accomplished by non-homologous end joining (NHEJ) or homologous recombination (HR) but replication-born DSBs are preferentially repaired by sister chromatid recombination (SCR), an HR reaction with the intact sister chromatid[6–8]. Alternatively, replication-born DSBs can be repaired by break-induced replication (BIR) or by HR with other homologous templates such as ectopic sequences or the homologous chromosome with the risks of causing a plethora of genetic instability phenotypes including loss of heterozygosity, deletions, insertions, translocations, or gross chromosomal rearrangements (GCRs)[9]. To favor SCR, sister chromatids are hold together by the cohesion ring[10], which is formed by two SMC (structural maintenance of chromosomes) components (Smc1 and Smc3) and two non-SMC components (Scc1 and Scc3). Sister chromatid cohesion at centromeres is essential and apparently sufficient for proper chromosome segregation in mitosis, but arm cohesion seems to have evolved to promote accurate post-replicative repair[11]. Indeed, although general cohesion is established after DNA replication, DNA damage triggers de novo cohesin loading facilitating repair[12–15]. The DNA damage response (DDR) promotes the establishment of such damage-induced cohesion at the break site as well as at other genomic regions independently of replication[15–17]. Damage-induced cohesion promotes SCR and is favored by certain factors such as the Smc5/6 complex[18,19], the constitutive methylation of H3K79 (ref. [20]), and by chromatin remodeling by the RSC complex[21].

In addition to cohesin proteins, several other factors have been described to contribute to the efficiency of SCR, such as the general HR factors Rad51, Rad52, Rad54, Rad59, Sae2, Sgs1, and Mus81, or the helicase Rrm3, which is part of the replisome[22–24]. The chromatin context at which the DSB occurs as well as the DSB-induced changes in chromatin can act as specific regulators of SCR. Thus, the higher H3K56 acetylation of newly synthesized DNA[25] seems to provide to sister chromatids a preference to be used as a substrate for replication-born DSBs repair versus non-sister templates[26]. However, the mechanism by which SCR is regulated versus other HR mechanisms is unknown.

We previously developed a yeast TINV-HO plasmid system based on a 24-bp mini-HO site (HOr) to induce ssDNA breaks that resulted in DSBs after replication and permitted the molecular analysis of their repair by SCR[6,10]. Since that system caused nicks on either DNA strand of the HOr site, we developed a system carrying the FRT recombination site of the yeast 2 μm circle and the inducible expression of a mutated form of the FLP endonuclease that permitted to induce strand-specific nicks at the FRT site. Using both sets of systems we have performed a detailed analysis of the impact of different chromatin factors, including remodelers and modifiers, on SCR. We identified Rpd3L and Hda1 histone deacetylases (HDACs) as specific regulators of SCR that prevent genetic instability. Interestingly, we show that both HDACs contribute to SCR by facilitating cohesin loading and promoting sister chromatid cohesion. These data allow us to propose a role for Hda1 and Rpd3L-mediated histone deacetylation in the maintenance of genome integrity by supporting

cohesion, thus favoring the repair of replication-born DSBs by SCR over other forms of DSB repair.

## Results

**Identification of chromatin factors affecting SCR.** To investigate the putative role of chromatin factors in the repair of replication-born DSBs, we analyzed unequal sister chromatid exchange (SCE), which is an accurate indicator of total SCR[6,10]. We built a plasmid containing both the previously reported TINV-HO system and the HO endonuclease gene under the control of the GAL1 galactose-inducible promoter and examined a selection of 27 mutants in chromatin remodelers and histone modifiers from the Euroscarf mutant collection. The TINV-HO system is based on two leu2 inverted repeats, one of which contains the HOr site, which is inefficiently targeted by the endonuclease HO leading mainly to DNA nicks that are converted into DSBs by the replication fork (Fig. 1a)[6,10]. DSB and SCE intermediates were detected by Southern blot analysis after wild type and mutant strains were induced to express the HO endonuclease (Fig. 1a, Supplementary Fig. 1a, see Methods). The quantification of the 4.7-kb SCE-specific band confirmed a 2–4-fold decrease in SCE levels in four control mutants, previously reported to be affected in SCR (dot1Δ, rad54Δ, rsc1Δ, and rsc2Δ). Five other mutants showed SCE levels below 10% (fun30Δ, swr1Δ, hda1Δ, rpd3Δ, and sap30Δ) whereas rph1Δ could not be analyzed due to its growth defect[27] (Fig. 1b). Among them, Sap30 and Rpd3 belong to the Rpd3L HDAC complex. Rpd3 is a class I HDAC that acts as the catalytic subunit and can also be part of another HDAC complex, Rpd3S. Both Rpd3L and Rpd3S complexes share the Sin3 and Ume1 subunits in addition to Rpd3 while they also contain specific subunits such as Sap30 or Rco1, which are specific for Rpd3L and Rpd3S, respectively[28]. We therefore added sap30Δ, rco1Δ, and sin3Δ for further analysis.

We next performed the same SCE analysis in a W303 genetic background (isogenic WS strains), which carries a deletion of the endogenous LEU2 gene to avoid interference with the leu2 repeats, and in which the HO endonuclease was expressed from the chromosome. The quantification of recombination intermediates after 3, 6, and 9 h of HO induction revealed that four of the mutants (fun30Δ, hda1Δ, sin3Δ, and sap30Δ) were strongly affected in the efficiency of SCE whereas swr1Δ and rph1Δ showed SCE levels similar to the wild type (Fig. 1c). Importantly, the percentage of HO-induced DSB was not significantly affected in any of the mutants but in sap30Δ, which grew more slowly and presented a delayed DSB induction with respect to the wild type (Supplementary Fig. 1b).

The TINV-HO system also allows studying genetically the appearance of Leu+ recombinants, which appear mostly by SCR, but that can be the consequence of other HR events. In agreement with a defect in SCR, rsc2Δ, hda1Δ, sin3Δ, and sap30Δ showed a significant decrease in the levels of HO-induced recombination (+HO) (Fig. 1d). We also detected decreased recombination in rph1Δ, suggesting that this mutant might be also affected in some other HR type of events. However, fun30Δ and swr1Δ efficiently repaired HO-induced breaks. The spontaneous (−HO) levels of recombination in these mutants were either not affected or higher than in the wild type (Fig. 1d). Higher levels of spontaneous inverted-repeats recombination in swr1Δ have been previously reported and are in agreement with a role for Swr1 in the maintenance of genetic stability[29]. By contrast, the increases observed in rsc2Δ, hda1Δ, and sap30Δ might reflect the incapacity of these strains to repair the replication-born DSBs via SCR (Fig. 1c), channeling spontaneous damage to other templates such as the other leu2 copy located in the same plasmid. Altogether, our screening results point to a specific role for Hda1 and Rpd3L,

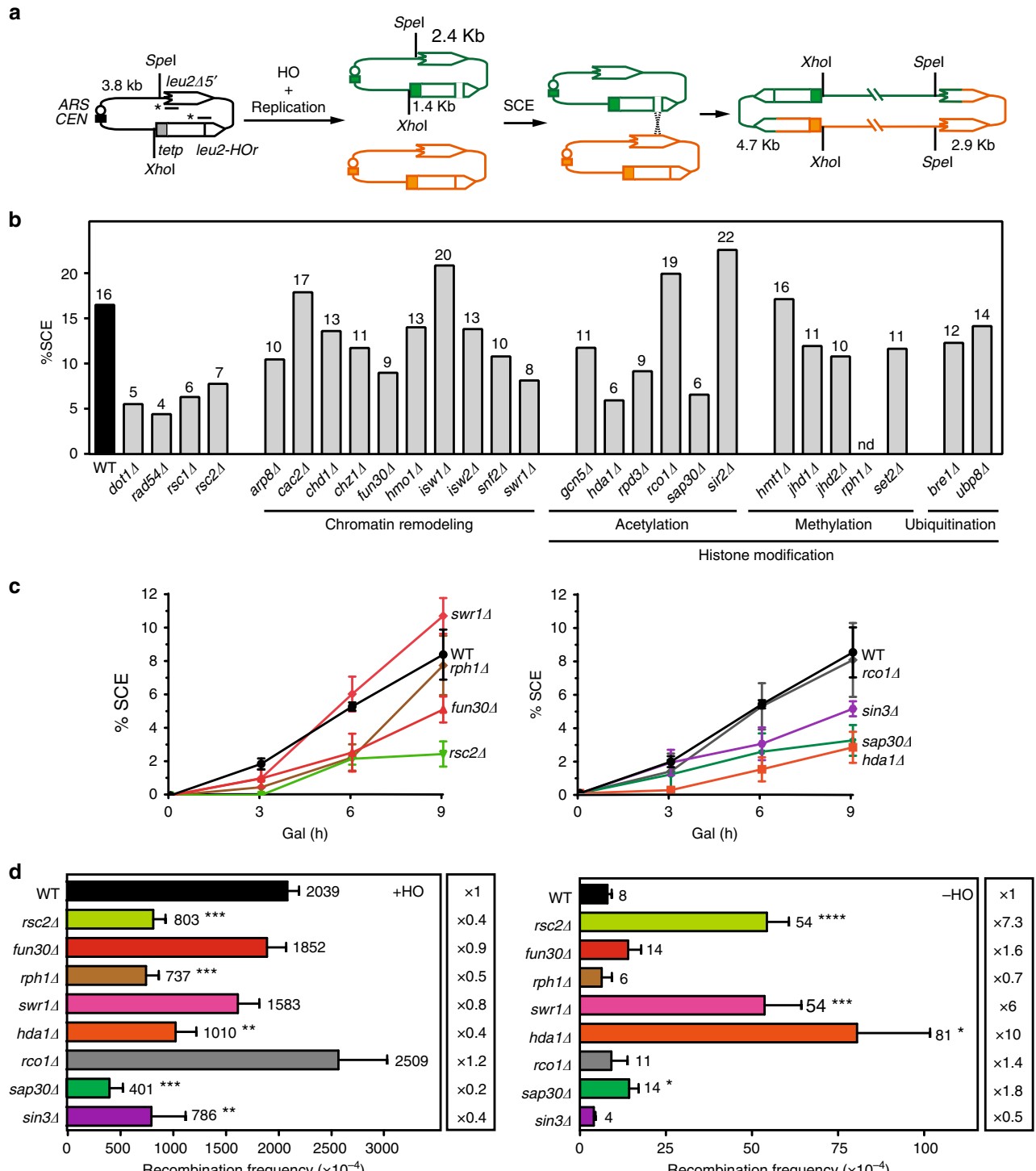

**Fig. 1** Screening for chromatin factors involved in SCR. **a** Schemes of the TINV-HO system and SCE intermediates produced after HO-induced replication-born DSBs. Fragments generated after *Xho*I–*Spe*I digestion are indicated with their corresponding sizes in kb and were detected by Southern blot hybridization with a *LEU2* probe (line with an asterisk). **b** Quantification of the SCE 4.7-kb fragment detected after 9 h of HO induction in wild type (BY4741) and the indicated mutant strains from the Euroscarf collection (see Supplementary Table 1) transformed with the pTHGH plasmid that contains both the TINV-HO system and GAL::HO ($n = 1$). **c** Quantification of the 4.7-kb SCE fragments during a time-course experiment after HO induction performed in wild type (WSR-7D), *rsc2Δ* (WSRSC2), *fun30Δ* (WSFUN30), *rph1Δ* (WSRPH1), *swr1Δ* (WSSWR1), *hda1Δ* (WSHDA1), *rco1Δ* (WSRCO1), *sap30Δ* (WSSAP30), and *sin3Δ* (WSSIN3) transformed with pRS316-TINV ($n \geq 2$). **d** Analysis of spontaneous (−HO) and HO-induced (+HO) recombination frequencies in wild type (WSR-7D), *rsc2Δ* (WSRSC2), *fun30Δ* (WSFUN30), *rph1Δ* (WSRPH1), *swr1Δ* (WSSWR1), *hda1Δ* (WSHDA1), *rco1Δ* (WSRCO1), *sap30Δ* (WSSAP30), and *sin3Δ* (WSSIN3) strains transformed with pRS316-TINV ($n \geq 3$). Means and SEM are plotted in **c** and **d**. *$p \leq 0.05$; **$p \leq 0.01$; ***$p \leq 0.001$; ****$p \leq 0.0001$ (two-tailed Student's *t*-test). See also Supplementary Fig. 1. nd, not determined. Data underlying this figure are provided as Source Data file

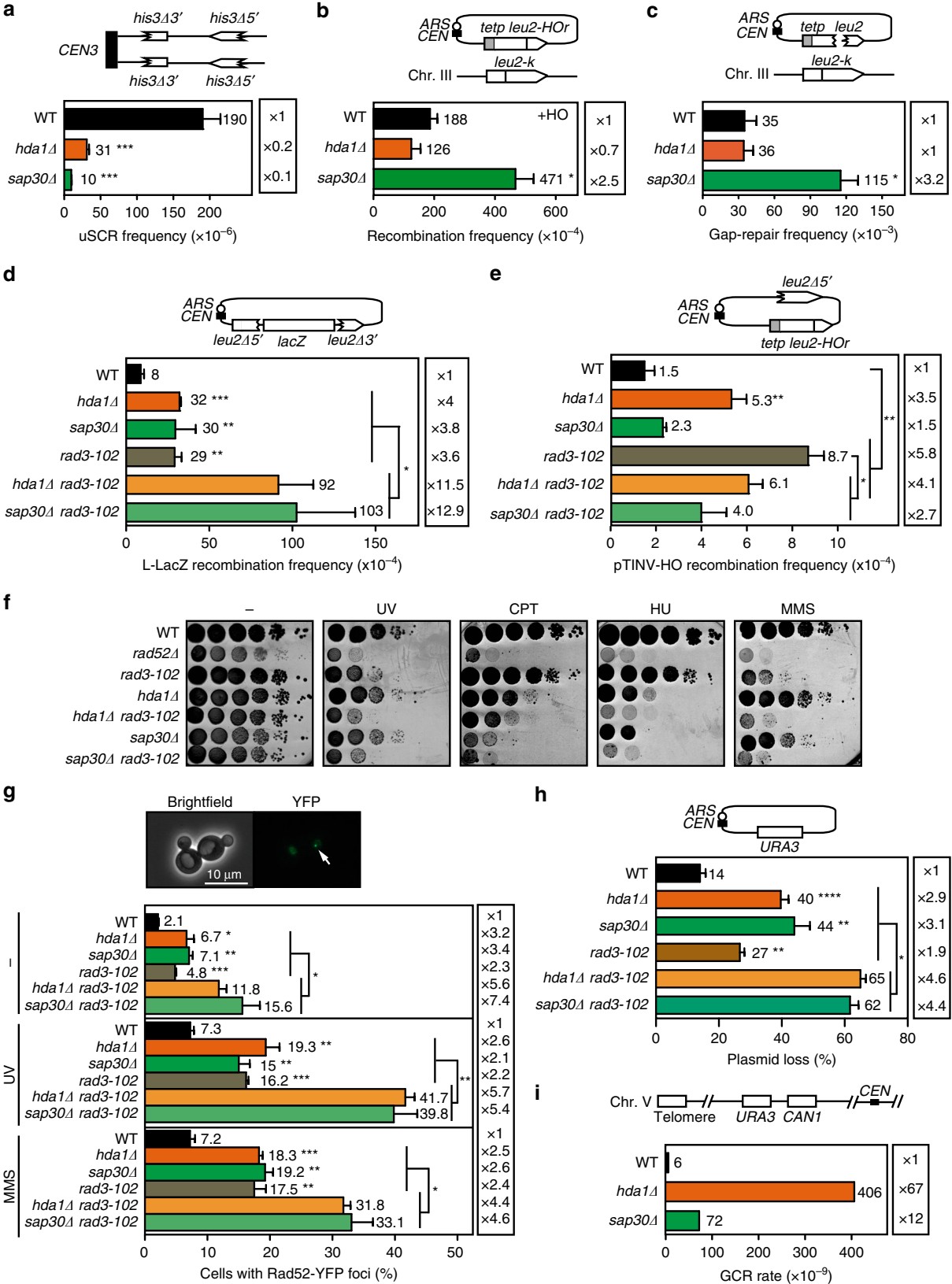

**Defective repair of replication-born DSBs in *hda1Δ* and *sap30Δ*.** We confirmed the SCR defect of *hda1Δ* and *sap30Δ* in a

different system located in a chromosome that measures spontaneous unequal SCR[30] (Fig. 2a) indicating that *hda1Δ* and *sap30Δ* affect the repair of not only HO-induced but also spontaneous DSBs, regardless of whether located in a plasmid or chromosome. By contrast, *hda1Δ* and *sap30Δ* showed no defect in the repair with non-sister templates, as shown for

but not Rpd3S, in the repair of replication-born DSBs using the sister chromatid as a template.

**Fig. 2** Defective repair of replication-born DSBs in *hda1Δ* and *sap30Δ*. **a** Analysis of spontaneous unequal SCR frequencies with the chromosomal direct-repeat *his3-Δ5'::his3-Δ3'* system in wild type (USCE), *hda1Δ* (USHDA1), and *sap30Δ* (USSAP30) strains ($n \geq 3$). **b** Analysis of HO-induced plasmid–chromosome recombination frequencies in wild type (W-Lk), *hda1Δ* (WLHDA1), and *sap30Δ* (WLSAP30) strains transformed with pCM189-L2HOr ($n = 3$). **c** Analysis of double-stranded DNA gap repair frequencies in wild type (W-Lk), *hda1Δ* (WLHDA1), and *sap30Δ* (WLSAP30) strains after transformation with *Mfe*I-digested pCM189-L2HOr or undigested pCM189-L2HOr ($n = 3$). **d** Analysis of direct-repeat recombination frequencies in wild type (WSRA), *hda1Δ* (WRHDA1), *sap30Δ* (WRSAP30), *rad3-102* (WRRAD3), *rad3-102 hda1Δ* (WRH1RD), and *rad3-102 sap30Δ* (WRS30RD) strains transformed with pSCH204 ($n = 3$). **e** Analysis of spontaneous recombination frequencies in wild type (WSRA), *hda1Δ* (WRHDA1), *sap30Δ* (WRSAP30), *rad3-102* (WRRAD3), *hda1Δ rad3-102* (WRH1RD), and *sap30Δ rad3-102* (WRS30RD) strains transformed with pRS316-TINV ($n \geq 3$). **f** Sensitivity to UV (40 J/m$^2$), CPT (20 μg/mL), HU (125 mM), and MMS (0.05%) of wild type (WSRA), *rad52Δ* (WS-52), *rad3-102* (WRRAD3), *hda1Δ* (WRHDA1), *hda1Δ rad3-102* (WRH1RD), *sap30Δ* (WRSAP30), and *sap30Δ rad3-102* (WRS30RD) strains. **g** Analysis of the percentage of S/G2 cells containing Rad52-YFP foci in wild type (WSRA), *rad3-102* (WRRAD3), *hda1Δ* (WRHDA1), *hda1Δ rad3-102* (WRH1RD), *sap30Δ* (WRSAP30), and *sap30Δ rad3-102* (WRS30RD) strains transformed with pWJ1344 in response to UV (10 J/m$^2$) or MMS (0.01%). A representative image of a cell with a Rad52-YFP foci is shown ($n = 3$). **h** Analysis of pRS316 plasmid loss in wild type (WSRA), *rad3-102* (WRRAD3), *hda1Δ* (WRHDA1), *hda1Δ rad3-102* (WRH1RD), *sap30Δ* (WRSAP30), and *sap30Δ rad3-102* (WRS30RD) strains ($n = 3$). **i** Analysis of the rate of gross chromosomal rearrangements (GCRs) at chromosome V in wild type (YKJM1), *hda1Δ* (YKHDA1), and *sap30Δ* (YKSAP30) strains ($n = 3$). A diagram of the different systems is depicted in **a–e**, **h** and **i**. Means and SEM are plotted in **a–e**, **g** and **h** whereas the mean is plotted in **i**. *$p \leq 0.05$; **$p \leq 0.01$; ***$p \leq 0.001$ (two-tailed Student's *t*-test). Data underlying this figure are provided as Source Data file

plasmid–chromosome recombination after a replication-born HO break or a double-stranded DNA gap[23], which is independent on the passage of the fork, implying that Hda1 and Sap30 are only required to repair replication-born DSBs with the sister chromatid (Fig. 2b, c). We rather observed that *sap30Δ*, but not *hda1Δ*, increased this type of ectopic recombination events suggesting that failures to repair with the sister chromatid could lead to channeling of the repair to non-sister templates. Furthermore, both *hda1Δ* and *sap30Δ* led to a significant increase in spontaneous direct-repeat recombination (*L-lacZ* direct-repeat system) (Fig. 2d) in agreement with some of the spontaneous damage being channeled to ectopic DNA copies.

We also studied SCE levels when, instead of using the HO endonuclease, nicks were originated by the Rad3 endonuclease. For this, we took advantage of the *rad3-102* allele, which impairs nucleotide excision repair (NER) postincision events leading to an accumulation of spontaneous ssDNA breaks[31]. *rad3-102* induction of nicks was reflected in an increase in the levels of Leu$^+$ recombinants with either the *L-lacZ* direct-repeat system (Fig. 2d) or the TINV-HO inverted-repeat system (Fig. 2e). These frequencies were significantly decreased in the absence of Hda1 or Sap30 in the TINV-HO system, which measures SCE, but enhanced in the *L-lacZ* system further in agreement with these proteins having a defect in SCE and channeling the repair to ectopic DNA sequences. Moreover, *hda1Δ rad3-102* and *sap30Δ rad3-102* were sensitive to UV, CPT, HU, and MMS, which enhance the possibilities of fork breakage (Fig. 2f), indicating that the defective repair of replication-induced damage leads to cell death. These results suggest that the loss of Hda1 and Sap30 could enhance genetic instability. To address this possibility, we assayed genetic instability at a more general scale by studying plasmid loss, Rad52 foci accumulation, and GCRs. In all three cases, we observed an increase in *hda1Δ* and *sap30Δ* mutants with respect to wild-type cells (Fig. 2g–i). In agreement with an increased occurrence of spontaneous unrepaired nicks, *rad3-102* enhanced the number of cells with Rad52 foci and the frequency of plasmid loss (Fig. 2g, h). These effects were significantly higher in *hda1Δ* and *sap30Δ* mutants. Such additive effects of *hda1Δ* and *sap30Δ* mutations over *rad3-102* on genetic instability were more evident after DNA damage, as observed in the frequency of cells with Rad52 foci after UV irradiation or MMS treatment (Fig. 2g). Altogether, our results support that *hda1Δ* and *sap30Δ* affect the repair of replication-born DSBs by the choice of a non-sister template leading to genetic instability.

**Deacetylation by Hda1 and Rpd3L is required for SCR.** Hda1 is a class II HDAC that acts as the putative catalytic subunit of the

Hda1 complex, and shares similarity with Rpd3 (ref. [32]). Therefore, it is possible that Hda1 and Rpd3 work through similar or common pathways. To assay this possibility, we studied the effect of the absence of both Rpd3L and Hda1 complexes. We repeated the SCE analysis with *sap30Δ*, *hda1Δ*, but now also including *rpd3Δ* and the double mutant *hda1Δ rpd3Δ*. Genetic analysis of HO-induced recombination (+HO) in the TINV-HO system revealed a similar decrease of around twofold in *rpd3Δ* and *hda1Δ rpd3Δ* mutants whereas spontaneous (−HO) recombination was increased to similar levels (Fig. 3a). Furthermore, no additive defect was observed in the *hda1Δ rpd3Δ* double mutant. We then performed a more complete time-point analysis of the repair reaction and changed our growing conditions from glycerol-lactate to raffinose to facilitate growth and DSB induction (Supplementary Fig. 2a). As shown in Fig. 3b, all mutants analyzed led to a similar decrease of around threefold in the efficiency of SCE respect to wild-type levels. These results suggest that Hda1 and Rpd3L HDAC complexes might act through a common pathway for SCR. We therefore continued with *rpd3Δ* to deepen into the possible mechanism behind the observed SCR defect.

To rule out any possible effects of Rpd3 in BIR, we used the previously reported system that measures the BIR intermediates between a full HO-cut chromosome XV and an intact chromosome VI by PCR[33]. We observed the same levels of BIR in wild type and *rpd3Δ* after 2, 4, and 6 h of HO induction revealing that Rpd3 is not involved in BIR (Fig. 3c). Therefore, Rpd3L is involved in SCR, and not in other replication-related recombination reactions such as BIR.

To test whether Rpd3 indeed acted through its HDAC function, we determined the effect of an Rpd3 deacetylase-dead mutant allele, *rpd3-H150A*[34]. Whereas the expression of wild-type Rpd3 (WT) complemented the SCE defect of *rpd3Δ* in both physical and genetic assays, the *rpd3-H150A* catalytically inactive allele was unable to rescue this phenotype and exhibited a strong SCE defect comparable to that of *rpd3Δ* cells transformed with an empty plasmid (Fig. 3d, e, Supplementary Fig. 2b). In agreement, spontaneous (−HO) recombination levels were enhanced by *rpd3-H150A* (Fig. 3e). This result implies that the deacetylase function of Rpd3 is required for efficient SCR.

**An FLP nickase-based system confirms the role for Rpd3 in SCR.** HO endonuclease induces mainly nicks at the *HOr* site in any of the DNA strands and can also occasionally target both strands at the same time, although the frequency of these replication-independent DSBs is very low (<2%)[10]. We therefore decided to create an improved version of the TINV-HO system in

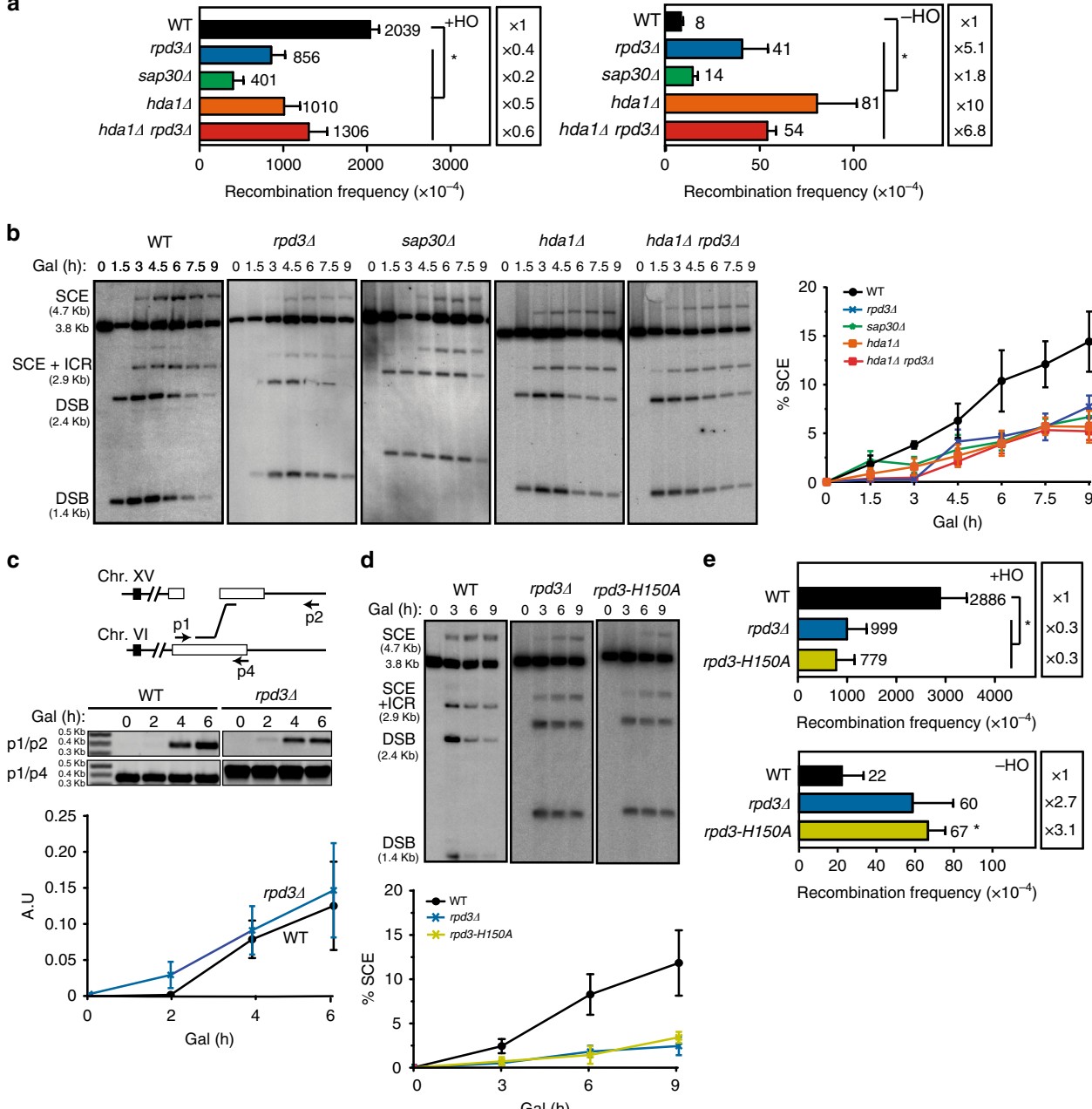

**Fig. 3** Deacetylation by Hda1 and Rpd3L impacts SCE through the same pathway. **a** Analysis of spontaneous (−HO) and HO-induced (+HO) recombination frequencies in wild type (WSR-7D), *rpd3Δ* (WSRPD3), *sap30Δ* (WSSAP30), *hda1Δ* (WSHDA1), and *hda1Δ rpd3Δ* (WSH1R3) strains transformed with pRS316-TINV (*n* ≥ 3). **b** Representative Southern blots and quantification of the 4.7-kb SCE fragments during time-course experiments after HO induction performed in wild type (WSR-7D), *rpd3Δ* (WSRPD3), *sap30Δ* (WSSAP30), *hda1Δ* (WSHDA1), and *hda1Δ rpd3Δ* (WSH1R3) strains transformed with pRS316-TINV (*n* ≥ 3). **c** Quantification of BIR intermediates in wild type (JFR-4) and *rpd3Δ* (JFRPD3) strains. A scheme of the BIR reaction is shown with the primers depicted as arrows. PCR with p1 and p2 oligos were used for the detection of BIR intermediates produced when the HO-cut chromosome III invades the homologous *ACT1* intron sequence on chromosome VI. PCR with p1 and p4 primers was used as a control. A representative image of the PCR products is shown (*n* = 4). **d** Representative Southern blots and quantification of the 4.7-kb SCE fragments during a time-course experiment after HO induction performed in the WSRPD3 (*rpd3Δ*) strain transformed with YEplac112-Rpd3 (WT), YEplac112 (*rpd3Δ*), or YEplac112-H150A (*rpd3-H150A*) plasmids (*n* ≥ 3). **e** Analysis of spontaneous (−HO) and HO-induced (+HO) recombination frequencies in the WSRPD3 (*rpd3Δ*) strain transformed with either YEplac112-Rpd3 (WT), YEplac112 (*rpd3Δ*), or YEplac112-H150A (*rpd3-H150A*) and the pRS316-TINV plasmids (*n* ≥ 4). Means and SEM are plotted in all panels. *\*p* ≤ 0.05 (two-tailed Student's *t*-test). In **a** and **d**, the 4.7-kb band is specific for SCE whereas the 2.9-kb band can result from SCE and other inter-chromatid recombination (ICR) events. The 3.8-kb band corresponds to the intact plasmid and 1.4 and 2.4-kb bands to the DSBs. See also Supplementary Fig. 2. Data underlying this figure are provided as Source Data file

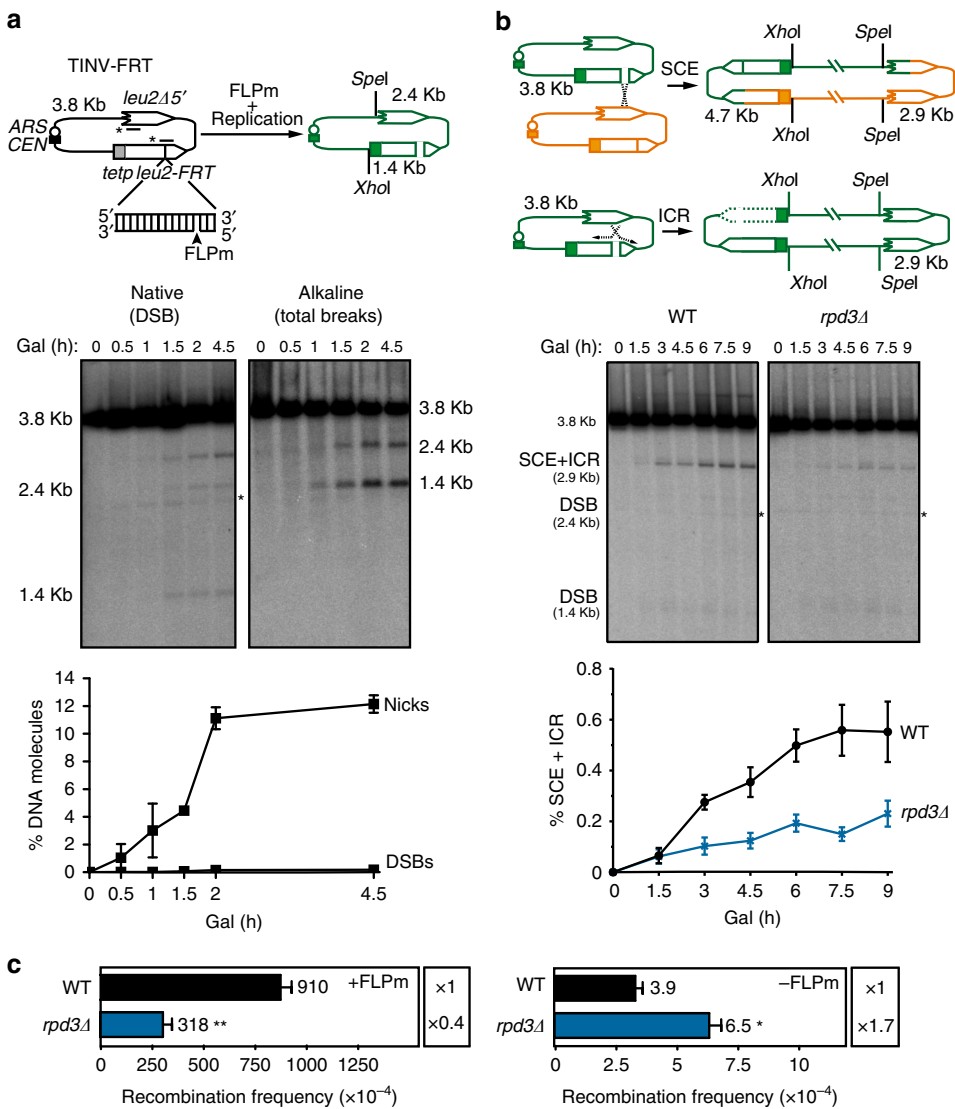

**Fig. 4** An FLP nickase-based system confirms the role for Rpd3 in SCR. **a** Analysis of the formation of nicks and DSBs after FLPm induction in the wild-type (WFLP) strain transformed with the pTINV-FRT plasmid. A scheme of the TINV-FRT system before and after the FLPm-induced nick is converted to a DSB by replication and representative Southern blots of the time-course analysis after FLPm induction after native and alkaline gel electrophoresis are shown ($n = 2$). **b** Representative Southern blots and quantification of the 2.9-kb SCE + ICR fragments during a time-course experiment after FLPm induction performed at 37 °C in wild type (WFLP) and $rpd3\Delta$ (WFRPD3) strains transformed with pTINV-FRT ($n \geq 3$). A scheme of the TINV-FRT system and SCE and ICR intermediates produced after FLPm-induced replication-born DSBs is shown. **c** Analysis of spontaneous (−FLPm) and FLPm-induced (+FLPm) recombination frequencies at 33 °C in wild type (WFLP) and $rpd3\Delta$ (WFRPD3) strains transformed with pTINV-FRT ($n = 3$). In **a** and **b**, fragments generated after $Xho$I–$Spe$I digestion are indicated with their corresponding sizes in kb and were detected by Southern blot hybridization with a $LEU2$ probe (line with an asterisk). Asterisks beside Southern blots indicate non-specific hybridization. Means and SEM are plotted in all panels. In **c**, *$p \leq 0.05$; **$p \leq 0.01$ (two-tailed Student's $t$-test). See also Supplementary Fig. 3A. Data underlying these figures are provided as Source Data file

which we introduced an FRT site instead of the $HOr$ site (TINV-FRT system). FRT is the target for the flipase recombinase (FLP), a mutated version of which (FLP-H305L, FLPm) causes an irreversibly protein-bound nick[35] and was previously used to study BIR[36]. Two hours after FLPm induction, 11% of the molecules were nicked, whereas less than 0.2% of molecules were cut in both strands, as we determined by the use of native and alkaline gel electrophoresis in which the 2.4 and 1.4-kb bands corresponded to either only DSBs in the native or to total breaks in the alkaline gel (Fig. 4a). These DSB levels were almost 20-fold lower than those obtained with the TINV-HO system, consistent with a notable improvement in the enrichment of replication-born DSBs versus other type of breaks. As a consequence of this decrease in the total percentage of DSBs, it was necessary to increase fourfold

the amount of DNA loaded on the gels and the strong signal of the 3.8-kb band corresponding to the fully linearized plasmid hindered the 4.7-kb SCE-specific band, impeding its reliable quantification. We therefore quantified the 2.9-kb band (SCE + ICR), mostly arising as a consequence of SCE intermediates but that can also reflect some intrachromatid recombination (ICR) events such as BIR intermediates[6]. As shown in Fig. 4b, the levels of SCE + ICR strongly decreased in the absence of Rpd3. As a consequence of repair, DSBs levels decreased after 3 h of FLPm induction in the wild type but they dropped more slowly in $rpd3\Delta$ cells (Supplementary Fig. 3a). When FLPm was induced, the genetic detection of Leu$^+$ recombinants increased by two orders of magnitude (Fig. 4c, +FLPm versus −FLPm) further confirming the formation of DSBs. FLPm-induced recombination levels

(+FLPm) dropped in *rpd3Δ* cells, but increased under spontaneous conditions (−FLPm). Altogether, these results validate the TINV-FRT as an improved system to measure the specific repair of replication-born DSBs through SCR and further confirm a role for Rpd3 in this process.

**Similar efficiency of SCR at the leading and lagging strands**. In addition to improving the specificity for nicks, the TINV-FRT system allows to direct the ssDNA break to either the Watson or the Crick strand, depending on the orientation of the FRT site. We therefore created two versions of the system to study the repair of nicks made in either the leading or the lagging strands by inserting FRT in each of the two opposite orientations (FRTd and FRTg, respectively) (Fig. 5a). Analysis of the DSBs' appearance and the efficiency of repair by using these two TINV-FRT systems revealed similar results for all time-points in wild-type cells (Supplementary Figs. 3b and 5a). This indicates that there are no major differences in the kinetics of SCE repair of DSBs arising as a consequence of a nick on either the leading or lagging strand.

In the genetic analysis, although the spontaneous recombination frequencies (−FLPm) were similar in both constructs ($17 \times 10^{-5}$ and $14 \times 10^{-5}$), the induction of the FLPm nickase (+FLPm) led to a twofold higher increase in the TINV-FRTd leading strand system (Fig. 5b). Since we detected no difference in the kinetics of SCE repair between both constructs (Fig. 5a), this difference must account for non-SCE recombination events. Importantly, *rpd3Δ* mutant led to a defective repair in both leading and lagging strands constructs as determined by both physical and genetic analyses (Fig. 5a, b), whereas spontaneous recombination levels were significantly higher (Fig. 5b), in agreement with our previous observations. Therefore, we conclude that DSBs arising as a consequence of forks that encounter a nick in either the leading or lagging strand are repaired by SCE with the same efficiency and require Rpd3.

**Rpd3 and Hda1 promote cohesin loading and chromatid cohesion**. We investigated the possible mechanisms by which Rpd3L could be required for efficient SCR. For this, we first performed a genetic analysis of the spontaneous and HO-induced recombination levels in double mutant strains carrying *rpd3Δ* in combination with mutations in genes encoding other factors previously reported to be involved in SCR (Rad51, Sae2, Sgs1, Mus81, Rrm3, Hst3, and Scc1). Double mutant combinations in *RPD3* and general HR factor genes (*RAD51, SAE2, SGS1, MUS81*) resulted in a further decrease in HO-induced recombination levels but no major changes in spontaneous recombination with respect to the single mutants (Fig. 6a, Supplementary Fig. 4a). This synergistic behavior points to a role for Rpd3 in SCR through a mechanism not related to general HR and is consistent with the spontaneous hyper-recombination phenotype observed (Fig. 2). By contrast, the levels of HO-induced recombination were significantly higher in *rpd3Δ rrm3Δ* and *rpd3Δ hst3Δ* than in either single mutant (Fig. 6a, upper panel), and the recombination frequency was already spontaneously augmented in *rpd3Δ rrm3Δ* (Supplementary Fig. 4a), likely reflecting higher levels of unrepaired damage. Instead, the combination of *rpd3Δ* with the *scc1-73* thermosensitive mutation, which impairs cohesion[37] and SCE[10], showed no significant changes in HO-induced or spontaneous recombination levels when assayed at the semipermissive temperature (33 °C) (Fig. 6a, lower panel). This suggests a possible epistatic relationship between these two factors. Genetic and physical analysis of FLPm-induced or spontaneous recombination showed analogous results, with the double *rpd3Δ*

*scc1-73* mutant presenting similar levels of FLPm-induced recombination frequency and SCE intermediates than either single mutant (Fig. 6b, c, Supplementary Fig. 4b, c). These results suggested a cohesin-related function as the responsible for the SCR defect observed in *rpd3Δ*.

Therefore, we analyzed cohesin levels in wild type and *rpd3Δ* cells by chromatin immunoprecipitation (ChIP) experiments of the MYC-tagged cohesin subunit Scc1 (Scc1-MYC). Given that cohesins are also recruited upon DSBs, we analyzed Scc1-MYC occupancy before and after HO induction (−HO and +HO, respectively) in the same sequences and conditions used to study SCR (Fig. 7a). Although cohesin loading was not significantly enhanced by HO induction, likely due to the low efficiency of HO cleaving the *HOr* site, we detected a threefold defect in cohesin loading before and after HO induction in the absence of Rpd3 (Fig. 7a) suggesting that *rpd3Δ* compromises cohesin loading regardless of DSB induction. We also monitored four regions of chromosome III, including *ARS305a*, *ARS305b*, *SYP1*, and the centromere (*CEN3*), prone to cohesin enrichment[38]. As shown in Fig. 7b, *rpd3Δ* caused a two- to threefold significant decrease of Scc1-MYC immunoprecipitation in all four regions analyzed in spontaneous conditions. Hence, we confirmed that Rpd3 loss causes a global defect in cohesin loading in undamaged conditions. Similar results were obtained after HO induction (Supplementary Fig. 5a).

We hypothesized that the lower loading of cohesins to chromatin should impact negatively on sister chromatid cohesion. Therefore, we monitored cohesion directly through a previously described system which is based on a *LacI::GFP* fusion and a multi-copy *LacO*-binding site placed at the *CEN3* (ref. [39]). In agreement with the presence of one single chromatid in G1, few cells presented two GFP foci in α-factor synchronized cells (Supplementary Fig. 5b, c). This percentage increased to 8 in G2/M in wild-type cells but to almost 23% in *rpd3Δ* cells indicating a clear defect in sister chromatid cohesion in the absence of Rpd3 (Fig. 7c). Importantly, and consistent with the epistatic relationship between *hda1Δ* and *rpd3Δ* regarding SCR, *hda1Δ* showed a similar defect in sister chromatid cohesion (Fig. 7c). However, when we tested other SCR-defective mutants such as *hst3Δ* and *rrm3Δ*, for which our double mutant analysis (Fig. 6) suggested that they act via a different mechanism, we only observed a slight but not significant decrease in either Scc1 occupancy (Fig. 7b) and sister chromatid cohesion (Fig. 7c). Further in agreement with the epistatic effects observed in recombination, the combination of Rpd3 loss with deletions in general HR factors (*sgs1Δ*) or specific replication/SCR factors (*rrm3Δ*) did not enhance this cohesion defect (Fig. 7c). Moreover, the loss of Rpd3 was epistatic with *scc1-73*, as expected from both factors affecting cohesion through a common pathway (Fig. 7c). To further determine whether the HDAC function of Rpd3 was behind its cohesion defect, we performed this assay with the *rpd3-H150/1A* deacetylase-dead mutant allele[40]. As observed for SCE (Fig. 3d, e), whereas the expression of wild-type Rpd3 (WT) complemented the cohesion defect of *rpd3Δ*, the catalytically inactive allele showed a cohesion defect similar to that of *rpd3Δ* cells transformed with an empty plasmid (Fig. 7d), implying that Rpd3 affects sister chromatid cohesion through its HDAC activity.

Finally, it is worth noting that Hda1 and Rpd3 have sequence similarity to Hos1 (ref. [32]), a deacetylase involved in Smc3 deacetylation after the S-phase to promote the separation of the sister chromatids[41,42]. Although Hda1 and Rpd3 have been reported not to be able to deacetylate Smc3 in vitro[41], our results could be explained if SCR required cohesin deacetylation and Hda1 and/or Rpd3 could act directly on cohesins

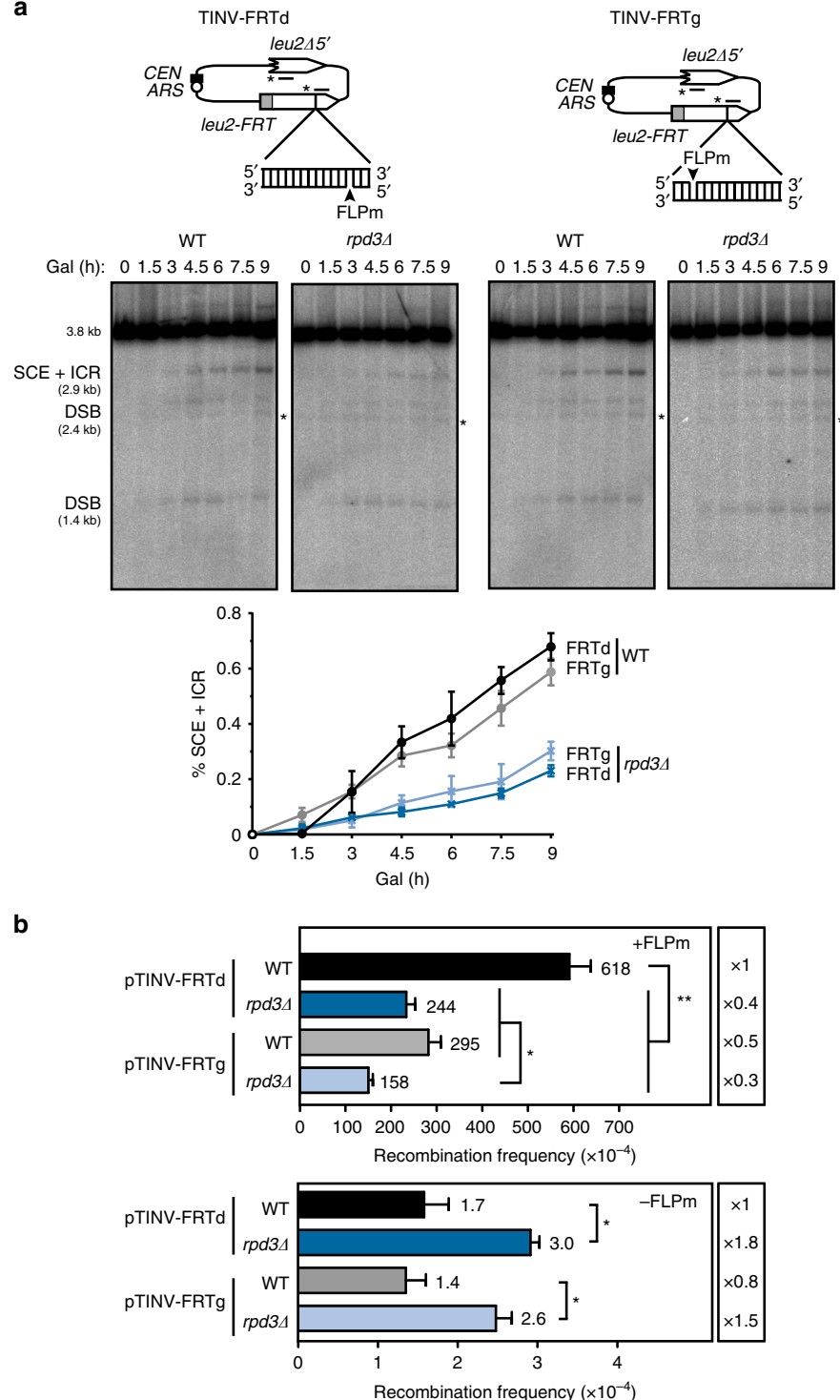

**Fig. 5** Similar efficiency of SCR at the leading and lagging strands. **a** Representative Southern blots and quantification of the 2.9-kb SCE + ICR fragments during a time-course experiment after FLPm induction in wild type (WFLP) and *rpd3Δ* (WFRPD3) strains transformed with pTINV-FRTd or pTINV-FRTg ($n = 3$). Schemes of the TINV-FRTd and TINV-FRTg systems are shown. Other details as in Fig. 4b. **b** Analysis of spontaneous (−FLPm) and FLPm-induced (+FLPm) recombination frequencies wild type (WFLP) and *rpd3Δ* (WFRPD3) strains transformed with pTINV-FRTd or pTINV-FRTg ($n = 3$). Means and SEM are plotted in all panels. In **b**, *$p \leq 0.05$; **$p \leq 0.01$ (two-tailed Student's *t*-test). See also Supplementary Fig. 3b. Data underlying this figure are provided as Source Data file

in vivo. If that were the case, we would expect that *hos1Δ* cells should have an SCE defect. As shown in Supplementary Fig. 5d, however, the efficiency of SCE was not affected in *hos1Δ* as it was in *rpd3Δ*, even though DSBs were induced with lower efficiency in *hos1Δ*. This result indicates that cohesin

deacetylation by itself is not required for efficient SCE, and support a role for the deacetylation of histones, rather than cohesins, in the repair of replication-born DSBs.

Altogether, our results demonstrate that Rpd3L and Hda1 are required for the efficient repair of replication-born DSBs through

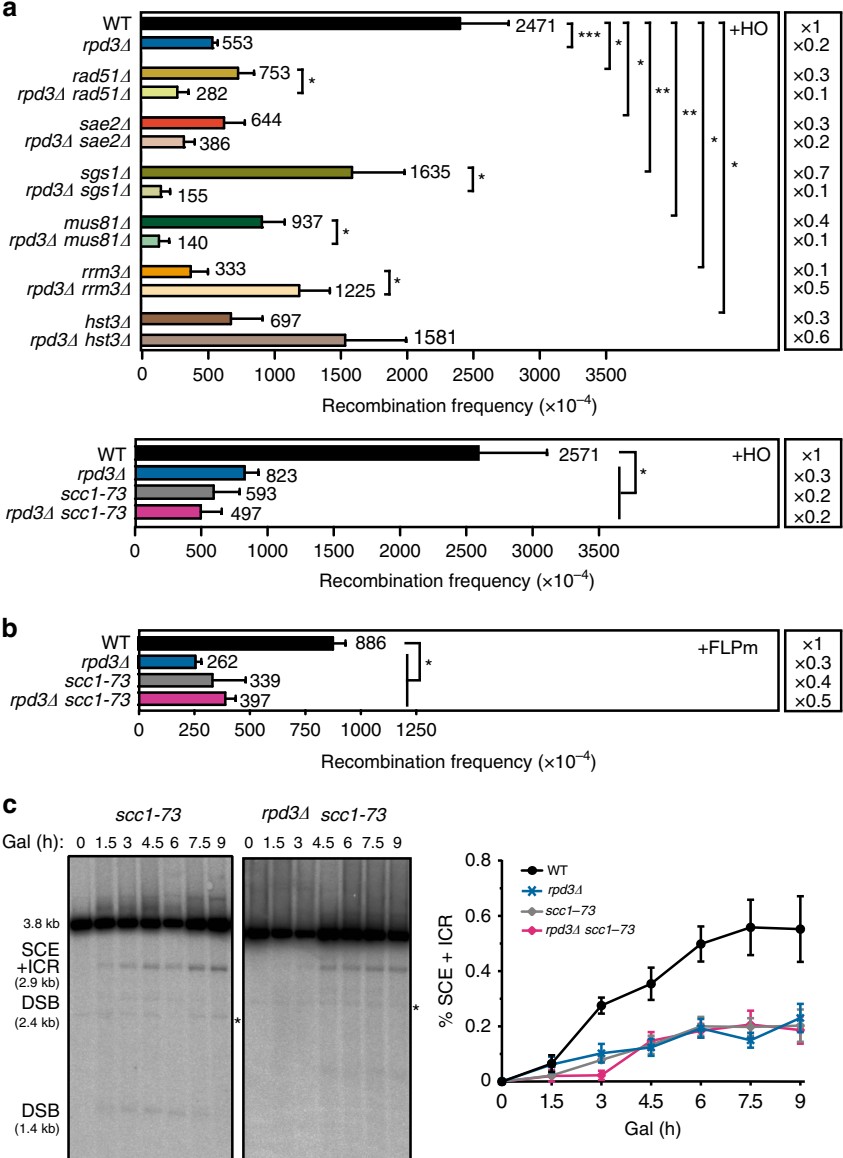

**Fig. 6** SCE levels in *rpd3Δ* in combination with other SCR-affected mutants. **a** Analysis of HO-induced recombination frequencies at 30 °C (upper panel) and 33 °C (lower panel) in wild type (WSR-7D), *rpd3Δ* (WSRPD3b), *rad51Δ* (WSRAD51), *rpd3Δ rad51Δ* (WSR3R5), *sae2Δ* (WSSAE2), *rpd3Δ sae2Δ* (WSR3S2), *sgs1Δ* (WSSGS1), *rpd3Δ sgs1Δ* (WSR3SG), *mus81Δ* (WSMUS81), *rpd3Δ mus81Δ* (WSR3M8), *rrm3Δ* (WSRRM3), *rpd3Δ rrm3Δ* (WSR3RR), *hst3Δ* (WSHST3), *rpd3Δ hst3Δ* (WSR3H3), *scc1-73* (WSSCC1), and *rpd3Δ scc1-73* (WSR3S1) strains transformed with pRS316-TINV (*n* ≥ 3). **b** Analysis of FLPm-induced recombination frequencies at 33 °C in wild type (WFLP), *rpd3Δ* (WFRPD3), *scc1-73* (WFSCC1), and *rpd3Δ scc1-73* (WFR3S1) strains transformed with pTINV-FRT (*n* = 3). **c** Representative Southern blots and quantification of the 2.9-kb SCE + ICR fragments during a time-course experiment after FLPm induction performed at 37 °C in wild type (WFLP), *rpd3Δ* (WFRPD3), *scc1-73* (WFSCC1), and *rpd3Δ scc1-73* (WFR3S1) strains transformed with pTINV-FRT (*n* ≥ 3). Other details as in Fig. 4b. Means and SEM are plotted in all panels. In **a** and **b** *$p ≤ 0.05$; **$p ≤ 0.01$ (two-tailed Student's *t*-test). See also Supplementary Fig. 4. Data underlying this figure are provided as Source Data file

SCR via a role for histone deacetylation in sister chromatid cohesion (Fig. 7e).

## Discussion

We show here an epigenetic regulation of the repair of replication-born DSBs by the modulation of sister chromatid cohesion. Through the use of a previously reported TINV-HO assay and an FRT/FLPm-based system for the study of repair of replication-born DSBs, we define a specific role of Rpd3L and HDACs in SCR. We demonstrate that this occurs through a role of these HDACs in sister chromatid cohesion by facilitating the general loading of cohesins (Fig. 7). Since cohesion between sister chromatids influences the efficiency of SCR, the major repair pathway for DSBs

arising at replication forks, the epigenetic regulation of cohesion has a strong impact on the repair of broken forks and the maintenance of genome integrity (Figs. 1–3).

The TINV-FRT system developed in this study strongly improved the efficiency of nicks versus unspeficic DSBs (Fig. 4a) and enabled to study repair of leading versus lagging strand forks (Fig. 5). We showed that nicks in the leading or lagging strand encountered by a fork converted into DSBs with a similar kinetics (Supplementary Fig. 3b) and were repaired with the same efficiency (Fig. 5). These results not only validate all our previous results obtained with the TINV-HO system in which nicks could occur randomly on either strand, but ratify TINV-FRT systems for the physical analysis of repair of replication-born DSBs by SCR.

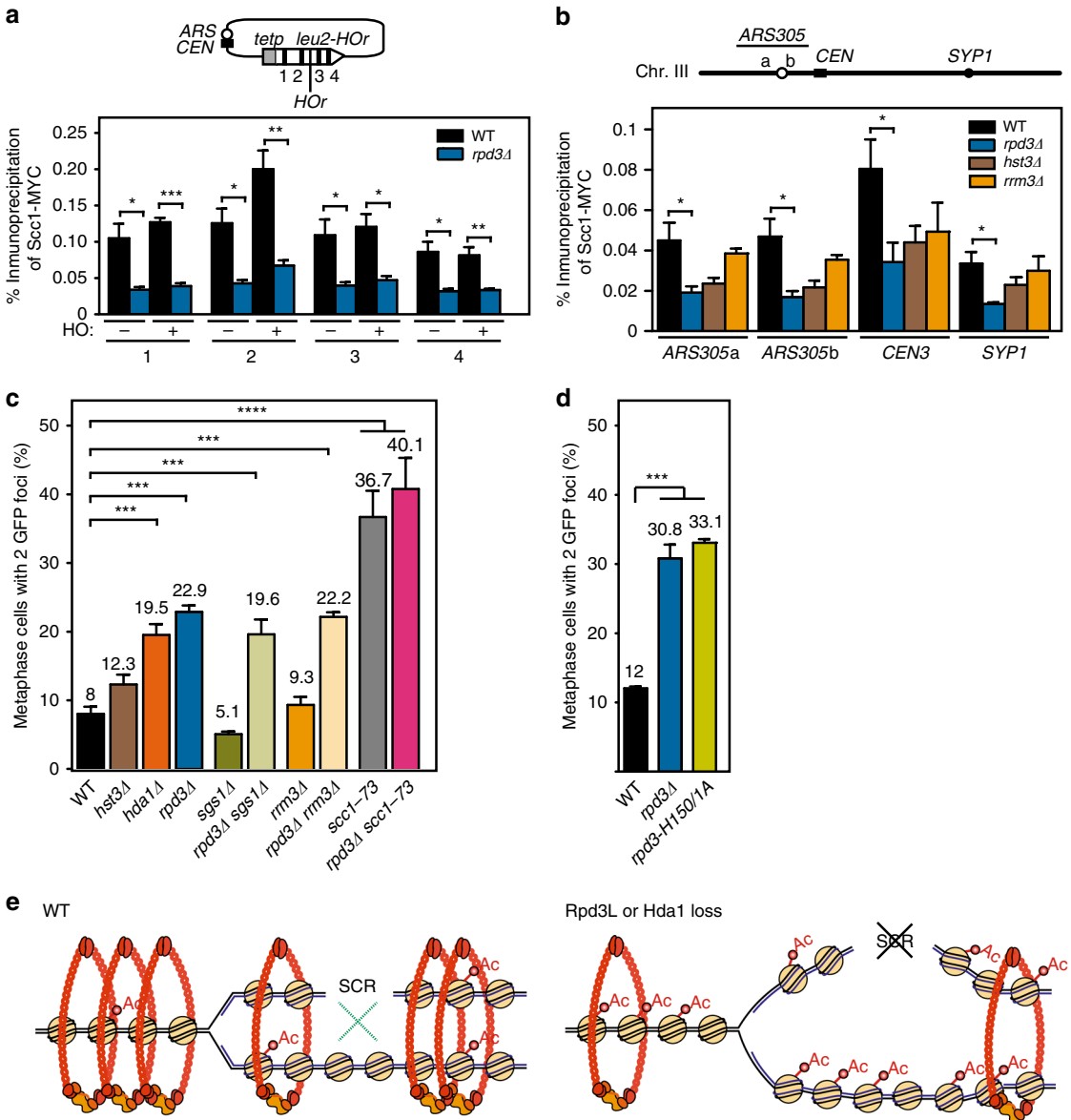

**Fig. 7** Rpd3 and Hda1 promote cohesin loading and sister chromatid cohesion. **a** ChIP analysis of Scc1-MYC occupancy in wild type (WSM) and rpd3Δ (WSMR3) strains after glucose (−HO) or galactose (+HO) addition for repression or activation of the HO endonuclease. A scheme of the different regions analyzed along the *leu2* gene of pCM189-L2HOr is depicted on top (n ≥ 3). **b** ChIP analysis of Scc1-MYC occupancy in wild type (WSM), rpd3Δ (WSMR3), hst3Δ (WSMH3), and rrm3Δ (WSMRR) strains. A scheme of the different chromosome III regions analyzed is depicted on top (n ≥ 3). **c** Percentage of G2 (nocodazol) arrested cells that have lost sister chromatid cohesion at centromere III, as indicated by the appearance of 2 GFP foci, in wild type (SBY885), hda1Δ (SBHDA1), rpd3Δ (SBRPD3), hst3Δ (SBHST3), sgs1Δ (SBSGS1), rpd3Δ sgs1Δ (SBR3SG), rrm3Δ (SBRRM3), rpd3Δ rrm3Δ (SBR3RR), scc1-73 (SBSCC1), and rpd3Δ scc1-73 (SBR3S1) (n ≥ 3). **d** Percentage of G2 (nocodazol) arrested cells that have lost sister chromatid cohesion at centromere III, as indicated by the appearance of 2 GFP foci, in SBRPD3 (rpd3Δ) strain transformed with PEN149 (WT), pRS315 (rpd3Δ), or PEN153 (rpd3-H150/1A) plasmids (n = 3). **e** A model to explain the role of Rpd3 in the repair of replication-born DSBs by SCR. In wild-type cells (WT), deacetylated chromatin supports cohesin loading, and thus sister chromatid cohesion, to favor SCR. By contrast, in the absence of Rpd3L or Hda1-mediated histone deacetylasaes, hyperacetylated chromatin would impair sister chromatid cohesion and hence affect the choice of the sister as a template for DSB repair. Means and SEM are plotted in **a**–**d**. In **a**–**d**, *p ≤ 0.05; **p ≤ 0.01; ***p ≤ 0.001 (two-tailed Student's t-test). See also Supplementary Fig. 5. Data underlying this figure are provided as Source Data file

Chromatin is a major determinant for all biological processes occurring on DNA. Here we define Rpd3L and Hda1-mediated histone deacetylation as a layer of regulation specific for SCR. Interestingly, Rpd3 and other HDACs are recruited to DSBs during HR repair already suggesting that they could have some roles in regulating HR[43]. We show that deacetylation is required for the choice of the sister chromatid to repair replication-born DNA breaks on either strand (Fig. 5). Indeed, the same SCR defect was observed in a deacetylase-dead mutant of Rpd3

(Fig. 3d, e). Therefore, histone deacetylation emerges as a major regulator of DSB repair in eukaryotes via SCR. Supporting the relevance of histone deacetylation and SCR in preventing genome instability, sap30Δ and hda1Δ mutants led to increased levels of unrepaired Rad52 foci, GCRs, and plasmid loss, and this was further enhanced when replication-born breaks were promoted by the *rad3-102* mutation (Fig. 2).

Importantly, the mechanism by which the loss of HDAC impairs SCR is linked to a general effect caused on cohesin

loading, which we observed both at the break site and in all other genomic regions analyzed (Fig. 7). Although an influence of de novo loading of cohesins to DSBs in SCR was previously reported[15,19–21], the fact that Rpd3L loss affected cohesin occupancy both at the break site as well as at the centromere and chromosome arms and in the absence of DSB induction (Fig. 7a, b) provides evidence for a role of general cohesin loading in SCR regardless of breakage. The defective cohesin loading has a direct impact on cohesion, as we demonstrate at the centromere in the absence of Rpd3 or Hda1 and in the catalytically inactive mutant of *RPD3* (Fig. 7c, d). This is unlikely related to a direct role on acetylated cohesins, as SCR is not impaired in the absence of the Hos1 cohesin deacetylase (Supplementary Fig. 5d). Hence, we support a model in which deacetylated chromatin stimulates cohesin loading, and thus sister chromatid cohesion, to favor SCR (Fig. 7e). By contrast, hyperacetylated chromatin would impair sister chromatid cohesion and affect the choice of the sister as a template to repair broken forks and likely any DSB occurring during S/G2. Chromatin appears, therefore, as a critical parameter for the stimulation of sister chromatid cohesion by controlling the levels of cohesins sitting on chromosomes.

In agreement with chromatin influencing cohesion, chromatin remodelers have been reported to influence cohesion[44–46]. Nonetheless, and although chromatin modification was hypothesized to impact cohesion[47], the only hint of a conection between chromatin modifications and sister chromatid cohesion comes from the observation that heterochromatin promotes cohesin association in fission yeast and higher eukaryotes[48–50]. Similarly, cohesion is enhanced in budding yeast heterochromatin-like silent chromatin, which is mediated by Sir2 class III HDAC, although this is independent on its HDAC function[51,52].

The connection between histone deacetylation and general cohesion uncovered here opens the possibility that other epigenetic marks could act at different levels in the regulation of cohesion and thus specifically impact SCR but not other mechanisms of DNA repair. Nonetheless, this does not exclude that other pathways beside sister chromatid cohesion could also impact the repair of broken forks by SCR. Indeed, a significant defect was not detected in *hst3Δ* or *rrm3Δ* mutants (Fig. 7), previously reported to specifically affect SCR, indicating that these two factors impact SCR likely through a different pathway. In this sense, it is worth mentioning that although a centromeric cohesion defect has been reported for *hst3Δ hst4Δ* double mutants, this was not observed in *hst3Δ* single mutants[39] in agreement with our results. Identifying how different factors control SCR besides sister chromatid cohesion should help us understand the mechanisms by which SCR is preferentially used over other types of homology-dependent DSB repair ptahways.

Hda1 and Rpd3L-deacetylated chromatin appears therefore as the milieu needed for the appropriate cohesion in the S-phase to promote SCR and ensure the accurate repair of broken forks. Interestingly, cohesins accumulate at stalled replication forks and seem required for efficient fork progression after DNA damage[53,54]. However, it is worthy to note that the action of Rpd3L was reported to be toxic in the S-phase checkpoint mutants[55]. Our results imply that although histone deacetylation can have deleterious consequences at stalled forks, it can also be beneficial for repair and fork progression and suggest that histone acetylation levels must be finely regulated to accomplish proper genome duplication.

Likely as a consequence of the innacurate repair of broken forks, we have observed several indicators of increased genetic instability in the absence of Hda1 and Rpd3L, such as ectopic recombination, plasmid loss, and GCRs (Fig. 2). Suggesting a possible conservation of this connection between histone acetylation and SCR in human cells, depletion of the human homologs of Sap30 and Sin3 also lead to increased DNA damage[56].

Interestingly, this damage was related to increased replication fork blockage caused by DNA–RNA hybrids[56]. It is thus possible that the Sin3A deacetylase complex also regulates repair of replication-born DSBs, whether or not induced by RNA–DNA hybrids, by promoting SCR through cohesion in human cells. Noteworthy, histone acetylation levels can impact the choice between HR and NHEJ pathways by influencing the chromatin association of the key repair factors BRCA1 and 53BP1 (ref. [57]) providing an example of the influence of chromatin structure in DSB repair pathway choice. Supporting that sister chromatid cohesion is essential for DNA repair also in higher eukaryotes, Scc1 depletion causes DNA damage sensitivity in DT40 and human cells[58–60]. Interestingly, ablation of cohesion in human cells was reported to cause GCRs as a consequence of the ligation of distal DNA breaks[61] and impaired cohesion was related to chromosomal instability and cancer[62]. These observations suggest that the regulation of SCR by cohesion might be conserved in higher eukaryotes and agree with our interpretation that the cohesion defects observed in the absence of Hda1 and Rpd3L are behind their genetic instability phenotypes.

Our study, therefore, not only uncovers a specific role of histone deacetylation in the repair of replication-born DSBs with the sister chromatid, but indicates a key role of histone acetylation levels in the loading of cohesins. The involvement of Hda1 and Rpd3 in cohesion might contribute to understand why patients with chromatin-related mutations show overlapping phenotypes with human diseases caused by mutations in cohesin-related genes (cohesinopaties), such as Cornelia de Lange syndrome[63]. Similarly, it is possible that some of the features of cohesinopaties are caused by defects in SCR or by the subsequent genetic instability. Strikingly, although cohesinopaties have not been related to cancer predisposition[64,65], cells from Cornelia de Lange patients show increased DNA damage sensitivity, particularly when cells are exposed to damage after replication[66]. Our study opens, thus, perspectives to understand the role of epigenetic modifications in the preservation of genome integrity and cancer prevention and ascertain the role of sister chromatid cohesion in promoting SCR as the most prominent DSB repair pathway.

## Methods

**Yeast strains and media**. Yeast strains used in this study are listed and described in Supplementary Table 1.

The WFLP strain was generated by *MET17* gene replacement with the GAL-FLPH250L::HPHMX6 fragment from pGAL-FLPH250L. pGALFLPH250L was previously built by inserting the *Pvu*II GAL-FLPH250L fragment from pBISGalkFLP[67] into *Pvu*II-digested pFA6aHPHMX6 (ref. [68]).

Media used in this study: YPAD (1% yeast extract, 2% bacto-peptone, 2% glucose, 20 mg/L adenine), SD (0.17% yeast nitrogen base (YNB) without amino acids nor ammoninum sulfate, 0.5% amounium sulfate and supplemented with amino acids. The absence of amino acid/s is specified when required), SC (SD containing 2% glucose). SGal (SD containing 2% filtered-galactose), SRaf (SD containing 2% raffinose). SGL (SD containing 3% filtered-glicerol and 2% sodium lactate), SPO (1% potassium acid, 0.1% yeast extract, 0.005% glucose), and FOA (S with half concentration of uracil (10 mg/L), 0.1 L-proline instead of amonion sulfate and 50 mg/L 5-FOA). Solid media were prepared adding 2% agar before autoclaving.

Yeast strains were freshly defrosted from stocks and grown at 30 °C, except for *scc1-73* strains that were grown at 26 °C, using standard practices. All experiments were performed at 30 °C unless specified.

**Plasmids**. Plasmids pCM189-L2HOr and pRS316-TINV[6] carrying the *leu2-HOr* allele, pWJ1344 carrying Rad52-YFP fusion[69], pSCH204 carrying the L-LacZ direct-repeat recombination system[70], pRS316, pFA6aKANMX4, pFA6aHPHMX4, and pFA6aNATNT2 used for gene replacement[68,71], YEplac112, YEplac112-Rpd3 and YEplac112-H150A[34], PEN149, pRS315, or PEN153 (ref. [40]) have been previously described.

pTHGH was generated by inserting the *Xma*I–*Sal*I GAL-HO fragment from pRS313GAL-HO[24] into pRS316-TINV[6]. pRS316-FRTa and pRS316-FRTb carrying the *leu2-FRT* allele with FRT site site in two orientations were generated by inserting the *Eco*RI-digested FRT sequence, obtained by primer (*Eco*RI-FRT-1 and *Eco*RI-FRT-2) annealing of FRT flanked with *Eco*RI restriction site sequences, into

*Eco*RI-digested pRS316-LEU2. Previously, pRS316-LEU2 was obtained by cloning a *Bam*HI–*Hind*III fragment from pCM189-LEU2 (refs. [6,72]) into pRS316. pTINV-FRT was constructed by substitution of *Bst*EII–*Hind*III fragment of pTINV-HO for *Bst*EII–*Hind*II fragment of pRS316-FRTa. pTINV-FRTb containing the *leu2-FRT* allele was constructed by substitution of *Bst*EII–*Hind*III fragment of pTINV-HO for *Bst*EII–*Hind*II fragment of pRS316-FRTb. pTINV-FRTd and pTINV-FRTg were generated by religation of *Pvu*II digested pTINV-FRT and pTINV-FRTb, respectively.

**Genotoxic damage sensitivity assay**. Mid-log cultures were grown in YPAD medium. Ten-fold dilutions of the culture prepared in sterile water were plated on solid YPAD medium containing the drugs at the indicated concentrations. UV irradiation was performed in the dried plates. Plates were incubated during 3 days (in the dark for UV-irradiated plates).

**Physical analysis of SCE intermediates**. Cells transformed with pTHGH, pRS316-TINV, pTINVFRT, pTINV-FRTd, or pTINVFRTg were grown to mid-log phase normally in SRaf (except in Fig. 1a–c, which were grown in SGL) and at 30 °C unless otherwise specified. In all cases, 5 μg/mL doxycycline was used to repress transcription from the *TET* promoter (*tetp*). Then, galactose (2%) was added to induce HO or FLPm expression. Samples were collected and DNA was extracted, digested with SpeI-XhoI (New England Biolabs), and analyzed by Southern blot hybridization using Hybond XL+ (GE Healthcare) membranes and detected with a [32]P-labeled 0.22-kb *LEU2* probe[73]. Original blots are provided as Source Data file. The *LEU2* probe was obtained by PCR using Leu2 Up 2000 and Leu Lo 2000 primers (Supplementary Table 2) and purified from agarose gels just before use. Quantification was performed by calculating the signal of the bands corresponding to DSBs, SCE, or SCE + ICR fragments relative to the total DNA in each line. For the analysis of nicks, DNA samples were electrophoresed at 4 °C in alkaline (50 mM NaOH, 1 mM EDTA) conditions. PhosphorImager Fujifilm FLA-5100 and the ImageGauge program were used for quantification.

**Genetic analysis of recombination**. Recombination frequencies were calculated as the median value of six independent colonies. The mean value of three independent transformants was plotted. For the LlacZ system, yeasts were grown in SC-trp plates and Leu+ recombinants were selected in SC-leu-trp. Recombination tests were performed with the TINV-HO or TINV-FRT systems. Briefly, mid-log phase cultures of yeast carrying the HO or FLPm gene under the control of the *GAL1* promoter were normally grown in SRaf (except in Figs. 1d and 3a, which were grown in SGL) liquid media with 5 μg/mL of doxycycline and at 30 °C unless indicated and split into two halves. One half was maintained in SRaf (or SGL in Figs. 1d and 3a) (−HO, −FLPm) and 2% of galactose was added to the other half (+HO, +FLPm) for 5 (HO) or 6 (FLP) hours. Leu+ recombinants were selected on SC-leu-ura.

**Chromosomal unequal SCR**. Chromosomal unequal SCR was assayed at the *his3Δ5′::his3Δ3′* system[30]. In this case, yeasts were grown in YPAD plates and His+ recombinants were selected on SC-his. The mean of three independent experiments was plotted.

**Double-stranded DNA gap repair**. The frequency of double-stranded DNA gap repair was calculated as the number of recombinants obtained after transformation with 200 ng of *Mfe*I-digested pCM189-L2HOr plasmid divided by the number of transformants obtained with 200 ng of undigested plasmid[23]. The mean of three independent experiments was plotted.

**Molecular detection of BIR intermediates**. For the detection of BIR intermediates, 2% galactose was added to induce HO expression to mid-log cultures of yeast strains containing the BIR assay grown in SRaf. DNA was then extracted and subjected to PCR reactions with different primers[33]. Original gels are provided as Source Data file. PhosphorImager Fujifilm FLA-5100 and the ImageGauge program were used for quantifying. The signal intensity of the PCR products amplified with p1 and p2 primers (Supplementary Table 2) was normalized to those of the PCR with p1 and p4 primers (Supplementary Table 2).

**Plasmid loss**. Colonies of independent transformants carrying the pRS316 plasmid were grown in YPAD plates for 2 days. Several dilutions were plated in YPAD (to score for total cells) and SC-trp (to score for cells which have lost the pRS316 plasmid). The frequency of plasmid loss was calculated as the median value of six independent colonies. The mean of three independent transformants was plotted.

**Gross chromosomal rearrangements**. Loss of *CAN1* and *URA3* marker were selected by plating late-log phase culture in SC-FOA with L-cannavaline (60 mg/L)[74]. The median of three independent experiments was plotted.

**Analyses of Rad52 foci**. Rad52 foci were counted in more than 200 S/G2 cells transformed with pWJ1344. For UV treatment, cells were resuspended in water onto Petri dishes as a 3-mm-deep cell suspension, UV-irradiated and incubated for 2 h before counting. For MMS treatment, MMS was added at the concentration indicated and incubated for 2 h before counting. Cells were visualized in Leica DC 350F. The mean and SEM of three different experiments was plotted.

**Chromosome immunprecipitation**. A yeast cell culture exponentially growing in SRaf media was split in two. One half was supplemented with 2% glucose and the other half with 2% galactose to induce HO induction (+HO) for 2 h. Cultures were crosslinked during 30 min with formaldehyde (1% final concetration). The reaction was stopped with glycine (125 mM final concentration). Samples were washed twice with PBS and collected into 1 mL tubes. Pellets were resuspended in lysis buffer (50 mM HEPES-KOH pH 7.5, 140 mM NaCl, 1 mM EDTA pH 8, Triton X-100, 0.1% sodium deoxycholate), complemented with protease inhibitors (cOmplete protease inhibitor cocktail (Roche) and 1 mM PMSF). Cell lysis was performed by shaking for 45 min with glass beads. Samples were collected and sonicated using Bioruptor (Diagenode), with cycles of 30 s during 30 min. Cell debris was eliminated by two rounds of centrifugation. Immunoprecipitation using Dynabeads Protein G (Invitrogen) for c-Myc monoclonal antibody (Clontech, cat no. 631206, 3:400) was carried out at 4 °C overnight and samples were washed four times with different solutions. Lysis buffer, lysis buffer complemented with 500 mM NaCl, a solution containing 10 mM Tris-HCl pH 8, 1 mM EDTA pH 8, 250 mM LiCl, 0.5% IGEPAL (Sigma), 0.5% SDS, and 0.5% sodium deoxycholate and last, a solution containing 10 mM Tris-HCl, 1 mM EDTA pH 8. Chromatin was eluted at 65 °C for 10 min with 150 μL of 50 mM Tris-HCl pH 7.4, 10 mM EDTA, 1% SDS, then treated with 6 μL of 50 mg/mL pronase at 42 °C for 2 h and then de-crosslinked for 6 h at 65 °C. DNA was cleaned up with a Quiagen purification kit. PCR primers used are shown in Supplementary Table 2.

**Sister chromatid cohesion assay**. SBY885 derivative strains (provided by Sue Biggins) were grown in YPAD to mid-log phase and supplemented with 25 μM CuSO4 to induce GFP expression. Then cells were G1-arrested with 2.5 μM of α-factor or G2/M-arrested with 15 mg/mL nocodazole. One milliliter of culture was fixed with 2.5% formaldehyde and visualized at the fluorescence microscope (Leica DC 350F). More than 200 cells were analyzed in each experiment.

**Reporting Summary**. Further information on research design is available in the Nature Research Reporting Summary linked to this article.

## Data Availability
Data underlying figures are provided as Source Data files. All data supporting the findings in the manuscript are available from the corresponding authors upon reasonable request.

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

## Acknowledgements

We thank Sue Biggins, Marc Gartenberg, Eulalia de Nadal, Kevin Struhl, and Jordi Torres-Rosell for kindly providing plasmids and strains. Research was supported by the European Research Council (ERC2014 AdG669898 TARLOOP), the Spanish Ministry of Economy and Competitiveness (BFU2016-75058-P), and the European Union (FEDER). P.O. was supported by a predoctoral training grant from the Spanish Ministry of Economy and Competitiveness and B.G.-G. by the Spanish Association Against Cancer (AECC).

## Author contributions

P.O. performed the experiments; P.O., B.G.-G. and A.A. designed the experiments and wrote the manuscript. All authors read, discussed, and agreed with the final version of this manuscript.

## Competing interests

The authors declare no competing interests.
