## [Peer Review File · Nature Communications]

Reviewers' comments:

Reviewer #1 (Remarks to the Author):

In this manuscript Ortega et al identify a role for histone deacetylation in the repair of replication-induced double strand breaks that are channelled through recombination between sister chromatids (SCR). The work provides an explanation for this role showing that cohesin complexes need the activity of histone deacetylases in order to load to the damaged sites and that favour recombinational repair between sister chromatids.

The manuscript starts with a systematic characterisation of what chromatin factors are involved in SCR using a previously published method that relies on a modified version of the HO site that leads to nicks (rather than DSBs) which are then converted to DSB during replication. Through a series of genetic tests and development of a second SCR assay based on the FLP, the authors arrive at the conclusion that HDAC complexes Rpd3L and Hda1 are specifically involved in repair of replication-induced breaks through SCR.

The authors then show that in the absence of HDACs, cohesins are not loaded properly to the sites of damage thus revealing the underlying reason for the defect in SCR observed. This therefore shows that damage-induced cohesion requires an environment with hypoacetylated histones.

The data overall are solid, the new assays are useful to the field and the authors clearly identify a role for HDACs in SCR through a requirement for cohesin loading which is the main claim of the manuscript.

It would be nice if the authors could show a role for HDACs in cohesin loading in undamaged conditions (i.e. during cohesion establishment/loop formation in cis). LA

Reviewer #2 (Remarks to the Author):

In this study, the authors address a role for histone deacetylases in fork-associated recombination through a mechanism involving cohesion and promoting sister chromatid cohesion (SCC). They demonstrate that the HDAC complexes, Rpd3L and Hda1 promote SCR using a number of reporter systems.

1. In the abstract it is stated: "Using these FLP-based systems and a previous one based on a mini-HO site, we analyzed a collection of yeast chromatin-modifying mutants to decipher the role of chromatin in eukaryotic DSB repair". This is true for the mini-HO system, but the collection was not analyzed with their new FLP-based system – A more accurate account of the work would include a revision of this sentence, as only *rad3Δ* is reported with the FLP system.

2. Page 7 (referring to Figure 2b)- " By contrast, *hda1Δ* and *sap30Δ* showed no defect in the repair of a double-stranded DNA gap, which is independent on the passage of the fork". The *sap30Δ* mutant is statistically different – what does this mean?

3. The interpretation of the drop assays in Figure 2f are somewhat for the various genotoxic agents (epistatic rather than additive). For example, 1. On CPT and MMS the *sap30Δ* and *sap30Δ rad3-102* look epistatic, 2. On HU, *rad3-102* and *hda1Δ rad3-102* look epistatic. This needs further interpretation.

4. To address the additive nature of 2d-2f in the context of DNA damage – measuring Rad52-YFP and GCR rate (or plasmid loss) in the double mutants in the presence of CPT, HU, and MMS would be more direct.

5. Figure 5d the quantification and gel are dissimilar for the *rad3-H150A*. The gel looks like more product forms in the *rad3-H150A* compared to *rad3Δ*. It also looks like in the *rad3-H150A* mutant

that DSB formation is not as effective.

6. Why did the authors switch to *rpd3Δ* (from *sap30Δ*) midway through their analysis. Was *sap30Δ* characterized in the FLP-based system?

7. What accounts for the difference in the spontaneous rates of recombination between the HO and FLP systems in the absence of GAL induction. +GAL the fold changes are similar.

-HO (fig 3b) -FLP (fig 4C)

WT 8 3.9

*rpd3Δ* 41 (X 5.1) 6.5 (X 1.7)

8. When drawing conclusions from the double mutant combinations in Figure 6a the levels should be compared (for statistical significant p value ) to *rpd3Δ* alone for the deletion of the HR factor genes *RAD51*, *SAE2*, *SGS1*, *MUS81*. In the case of *sae2Δ* and *rpd3Δ* *sae2Δ* is there a difference? The epistatic vs additive argumentation is difficult to understand. *rpd3Δ* (521) is 1.8 and 1.3 above *rpd3Δ rad51Δ* (282) and *rpd3Δ sae2Δ* (386) respectively and called additive, whereas with *scc3-1* at 33C, *rpd3Δ* (1017) is 2.0 X above *rpd3Δ scc1-73* and considered epistatic?

9. Figure 7b for the recovery of *Scc1-Myc* on the plasmid (which is not HO dependent). To interpret these data most effectively it would be beneficial to add back the WT -*RPD3* and the *rpd3-H150A* and measure cohesion recovery.

10. To address additive vs epistatic in SCR and cohesion and correlate these events, measuring cohesion in double mutants will help integrate the results of Figures 6 and 7. Measuring SCC, like in Figure 7C, in double mutants *rpd3 sgs1* *rpd3 rrm3* and *rpd3 scc1-73* and relative to single mutants would tie data together.

11. Figure 7d would be better in Supp.

12. The discussion should be streamlined and interact with the results - pointing out the really significant advancements the work makes

REBUTTAL

Reviewer #1 (Remarks to the Author):

In this manuscript Ortega et al identify a role for histone deacetylation in the repair of replication-induced double strand breaks that are channelled through recombination between sister chromatids (SCR). The work provides an explanation for this role showing that cohesin complexes need the activity of histone deacetylases in order to load to the damaged sites and that favour recombinational repair between sister chromatids.

The manuscript starts with a systematic characterisation of what chromatin factors are involved in SCR using a previously published method that relies on a modified version of the HO site that leads to nicks (rather than DSBs) which are then converted to DSB during replication. Through a series of genetic tests and development of a second SCR assay based on the FLP, the authors arrive at the conclusion that HDAC complexes Rpd3L and Hda1 are specifically involved in repair of replication-induced breaks through SCR.

The authors then show that in the absence of HDACs, cohesins are not loaded properly to the sites of damage thus revealing the underlying reason for the defect in SCR observed. This therefore shows that damage-induce cohesion requires an environment with hypoacetylated histones.

The data overall are solid, the new assays are useful to the field and the authors clearly identify a role for HDACs in SCR through a requirement for cohesin loading which is the main claim of the manuscript.

It would be nice if the authors could show a role for HDACs in cohesin loading in undamaged conditions (i.e. during cohesion establishment/loop formation in cis). LA

Thanks for all constructive comments.

As requested, we have now included data for cohesin loading in both HO-induced and undamaged conditions along chromosome III (Figure S5A and 7B, respectively) concluding that Rpd3 loss causes a defect in sister chromatid cohesion regardless of DNA damage. This is also concluded from figure 7A (performed after HO-induction or in undamaged conditions) and 7B and C, both of which show data from experiments in undamaged conditions.

Reviewer #2 (Remarks to the Author):

In this study, the authors address a role for histone deacetylases in fork-associated recombination through a mechanism involving cohesion and promoting sister chromatid cohesion (SCC). They demonstrate that the HDAC complexes, Rpd3L and Hda1 promote SCR using a number of reporter systems.

1. In the abstract it is stated: "Using these FLP-based systems and a previous one based on a mini-HO site, we analyzed a collection of yeast chromatin-modifying mutants to decipher the role of chromatin in eukaryotic DSB repair". This is true for the mini-HO system, but the collection was not analyzed with their new FLP-based system –A more accurate account of the work would include a revision of this sentence, as only *rpd3Δ* is reported with the FLP system.

The abstract has been modified accordingly. Thank you.

2. Page 7 (referring to Figure 2b)- " By contrast, *hda1Δ* and *sap30Δ* showed no defect in the repair of a double-stranded DNA gap, which is independent on the passage of the fork". The *sap30Δ* mutant is statistically different – what does this mean?

Both gap-repair and plasmid chromosome recombination assays revealed an increased frequency in *sap30Δ*, but not in *hda1Δ*. However, both *hda1Δ* and *sap30Δ* showed increased direct-repeat recombination frequencies. This could be due to some role of Sap30 in NHEJ (an Rpd3 role in NHEJ has been reported by Tao et al., *Cell Res.* 2011). In the absence of Sap30, defects in NHEJ could favor HR. Alternatively, the increased recombination frequencies of *sap30Δ* in our systems could reflect a role of Sap30 in DSB resection or heteroduplex extension. Short events would favor *Leu+* gene conversion recombinant products, whereas long events would reach the *leu2-k* mutation producing no *Leu+* recombinants. However, we find the effect being specific for SCR but not for plasmid-chromosome recombination; therefore, it is unlikely that a general defect in resection or heteroduplex extension explain the difference. Since this is out of the scope of this manuscript we have preferred not to include it in the discussion, as it would unnecessarily make the manuscript more difficult to read.

3. The interpretation of the drop assays in Figure 2f are somewhat for the various genotoxic agents (epistatic rather than additive). For example, 1. On CPT and MMS the *sap30Δ* and *sap30Δ rad3-102* look epistatic, 2. On HU, *rad3-102* and *hda1Δ rad3-102* look epistatic. This needs further interpretation.

We agree that the drop assays shown were not easy to interpret due to loading controls (YPAD plate

without treatments). We have therefore repeated them with better loading controls, which now include a similar number of cells for each of the strains. We believe that the additive effects are now evident and much clear. Thank you.

4. To address the additive nature of 2d-2f in the context of DNA damage – measuring Rad52-YFP and GCR rate (or plasmid loss) in the double mutants in the presence of CPT, HU, and MMS would be more direct.

We have now included data for Rad52 foci and plasmid loss in the double mutants (figures 2G and H) as well as the effect of UV and MMS. The new data clearly agree with an additive effect for *rad3-102* and *had1Δ* or *rpd3Δ* both in spontaneous and damaged conditions. Thank you.

5. Figure 5d the quantification and gel are dissimilar for the *rpd3-H150A*. The gel looks like more product forms in the *rpd3-H150A* compared to *rpd3Δ*. It also looks like in the *rpd3-H150A* mutant that DSB formation is not as effective.

A new experiment was added and a more representative gel picture of the average data is shown. Thank you.

6. Why did the authors switch to *rpd3Δ* (from *sap30Δ*) midway through their analysis. Was *sap30Δ* characterized in the FLP-based system?

We initially analyzed *Sap30* as an *Rpd3L*-specific factor versus *Rco1* (*Rpd3S*-specific factor). However, once we saw that *Rpd3S* loss had no effect and that *Hda1* and *Rpd3* were additive, we decided to continue our experiments with *rpd3Δ*, given that *Rpd3* is the catalytic subunit of the *Rpd3L* complex and that the *rpd3Δ* mutant grows better than *sap30Δ*. We have clarified this now in the text.

7. What accounts for the difference in the spontaneous rates of recombination between the HO and FLP systems in the absence of GAL induction. +GAL the fold changes are similar.

-HO (fig 3b) -FLP (fig 4C)

WT 8 3.9

*rpd3Δ* 41 (X 5.1) 6.5 (X 1.7)

These differences are due to the use of glycerol lactate or raffinose media. Figure 3B (now 3A) represents values in glycerol lactate media, whereas Figure 4C represents values in raffinose media. This is now more clearly stated in the text and methods sections.

8. When drawing conclusions from the double mutant combinations in Figure 6a the levels should be compared (for statistical significant p value) to *rpd3Δ* alone for the deletion of the HR factor genes *RAD51*, *SAE2*, *SGS1*, *MUS81*. In the case of *sae2Δ* and *rpd3Δ sae2Δ* is there a difference? The epistatic vs additive argumentation is difficult to understand. *rpd3Δ* (521) is 1.8 and 1.3 above *rpd3Δ rad51Δ* (282) and *rpd3Δ sae2Δ* (386) respectively and called additive, whereas with *scc3-1* at 33C, *rpd3Δ* (1017) is 2.0 X above *rpd3Δ scc1-73* and considered epistatic?

We have repeated some of the tests with *rpd3Δ* and the statistic significances of all the comparisons are now shown. All data of the double mutants is different to one or both of their respective single mutants except in the case of *rpd3Δ scc1-73*. This suggests a possible epistatic relationship between *Rpd3* and *Scc1*, which we then decided to study in more detail. New data are incorporated in new Fig. 6A

9. Figure 7b for the recovery of *Scc1-Myc* on the plasmid (which is not HO dependent). To interpret these data most effectively it would be beneficial to add back the WT -*RPD3* and the *rpd3-H150A* and measure cohesion recovery.

Thanks for the suggestion. We have performed this experiment and the results are as expected. This is now discussed in the text (page 16) and shown in new Figure 7D.

10. To address additive vs epistatic in SCR and cohesion and correlate these events, measuring cohesion in double mutants will help integrate the results of Figures 6 and 7. Measuring SCC, like in Figure 7C, in double mutants *rpd3 sgs1 rpd3 rrm3* and *rpd3 scc1-73* and relative to single mutants would tie data together.

We have now performed cohesion experiments in the double mutants as suggested (data in new figure 7C and text in page 16) with the expected results. Thank you.

11. Figure 7d would be better in Supp.

Moved to Figure S5D.

12. The discussion should be streamlined and interact with the results - pointing out the really significant advancements the work makes.

We have tried to improve the Discussion making emphasis on the significant advances, as suggested. Thanks very much for helping make the manuscript stronger and conclusions clear.

REVIEWERS' COMMENTS:

Reviewer #1 (Remarks to the Author):

The authors have addressed my previous comments. This is a nice piece of work and I fully support its publication. LA.

Reviewer #2 (Remarks to the Author):

The authors have address my concerns.

The quality and integration of the data within the manuscript have been improved. The work contributes to the field.

REVIEWERS' COMMENTS:

Reviewer #1 (Remarks to the Author):

The authors have addressed my previous comments. This is a nice piece of work and I fully support its publication. LA.

Thank you for the revision and the positive comments.

Reviewer #2 (Remarks to the Author):

The authors have address my concerns.

The quality and integration of theie data within the manuscript have been improved. The work contributes to the field.

Thank you for the revision and the positive comments.